# Early Life Determinants of Overweight and Obesity in a Sample of Mexico City Preschoolers

**DOI:** 10.3390/nu17040697

**Published:** 2025-02-15

**Authors:** Reyna Sámano, Salma Lopezmalo-Casares, Hugo Martínez-Rojano, Gabriela Chico-Barba, Ricardo Gamboa, Estibeyesbo Said Plascencia-Nieto, Ashley Diaz-Medina, Cristina Rodríguez-Marquez, María Elena Téllez-Villagómez

**Affiliations:** 1Coordinación de Nutrición y Bioprogramación, Instituto Nacional de Perinatología, Secretaría de Salud Montes Urales 800, Lomas de Virreyes, Alcaldía Miguel Hidalgo, Ciudad de México 11000, Mexico; gabyc3@gmail.com; 2Escuela de Dietética y Nutrición del ISSSTE, Callejón Vía, Av. San Fernando No. 12, San Pedro Apóstol, Tlalpan, Ciudad de México 14070, Mexico; salma.lopezmalo@ednissste.com.mx (S.L.-C.); mariacris.rodriguez@issste.gob.mx (C.R.-M.); maria.tellez@issste.gob.mx (M.E.T.-V.); 3Departamento de Posgrado e Investigación, Escuela Superior de Medicina del Instituto Politécnico Nacional, Plan de San Luis y Díaz Mirón s/n, Colonia Casco de Santo Tomas, Delegación Miguel Hidalgo, Ciudad de México 11340, Mexico; saidpn@yahoo.com.mx; 4Departamento de Fisiología, Instituto Nacional de Cardiología “Ignacio Chávez”, Ciudad de México 14080, Mexico; rgamboaa_2000@yahoo.com; 5Programa de Maestría en Ciencias de la Salud, Escuela Superior de Medicina del Instituto Politécnico Nacional, Plan de San Luis y Díaz Mirón s/n, Colonia Casco de Santo Tomas, Delegación Miguel Hidalgo, Ciudad de México 11340, Mexico; d.ash_a@ymail.com

**Keywords:** breastfeeding, exclusive breastfeeding, body mass index, preschooler, pediatric obesity, nutritional status, Mexico

## Abstract

Introduction: Childhood obesity is a growing public health problem with long-term consequences. Understanding the early contributing factors is crucial for prevention and early intervention. This study explored the influence of breastfeeding, birth weight, gestational age, parental education, and sex on body mass index (BMI) during infancy. Methods: Standardized weight and height measurements of children followed a common World Health Organization protocol. Information on sex, gestational age, birth weight, breastfeeding practices and duration, family income, and mother’s educational level, as well as other sociodemographic factors, was collected from clinical records. Linear regression models were calculated. Results: This study analyzed factors associated with overweight and obesity in 286 children under 5 years of age, using data from daycare records. Several significant associations were found. Regarding breastfeeding, while 85% of children received breast milk, only 23% did so exclusively for at least six months. Although no significant difference was observed in BMI change between exclusive and partial breastfeeding groups between birth and 5 years of age, the duration of exclusive breastfeeding, the birth BMI, and the educational level predicted 54% of the variability in BMI percentile change from birth to two years (*p* = 0.001). In addition, girls showed significantly longer exclusive breastfeeding. Regarding gestational age, preterm infants showed a significantly greater increase in BMI percentile compared to term infants. Gestational age also proved to be a significant factor in explaining BMI variability up to 5 years of age. Regarding sex, at age 5, boys showed a significantly higher prevalence of overweight and obesity than girls. With respect to family income, no statistically significant difference was found in BMI change between birth and 2 years of age; however, this variable warrants further investigation in future studies with greater statistical power. Finally, birth BMI was a significant predictor of BMI variability at 5 years of age. Conclusions: In this study, gestational age, sex, birth BMI, and the duration of exclusive breastfeeding were the most important determinants of BMI and the prevalence of overweight and obesity in children up to 5 years of age. Further studies are needed to thoroughly explore the role of family income and other factors.

## 1. Introduction

Childhood obesity, a growing public health problem primarily affecting low- and middle-income countries (80% of cases in 2020) [1], is a disease in itself and a risk factor for non-communicable chronic diseases. A significant increase is projected for 2035: more than 750 million children (aged 5 to 19), two out of five worldwide, will be overweight or obese, mainly in middle-income countries [2,3]. More than 60% of children who are overweight before puberty will remain so in early adulthood [4], with a projected increase from more than 430 million (22% of the global population aged 5 to 19 in 2000–2016) to 770 million (39%) in 2035 [3,5]. Without interventions, it is estimated that by 2035, 27 million children will suffer from hyperglycemia, 68 million from hypertension, and 76 million will have low HDL cholesterol levels, attributable to a high body mass index (BMI).

This problem is especially serious in Mexico, where a severe epidemic of childhood overweight and obesity is observed, with a progressive increase in prevalence over the last three decades, according to the National Health and Nutrition Surveys (ENSANUT) [6]. ENSANUT 2016 reported a six-fold increase in the prevalence of overweight and obesity (from 5.3% in children under 5 years of age to 31.4% in schoolchildren aged 5 to 11), while ENSANUT 2018 shows a 4.3-fold increase (from 8.2% in children aged 0 to 4 years to 35.6% in schoolchildren) [7]. This high prevalence in children under 11 years of age (almost half the peak observed in adults) is a serious concern for child health [7,8].

Several early-life factors influence the development of obesity by affecting body composition and appetite [9]. Birth weight, reflecting intrauterine nutrition [10], is an important perinatal factor [11]. However, the relationship between birth weight, prematurity, sex, socioeconomic level, parental educational level, and adolescent obesity, as well as the relationship between breastfeeding and the addition of complementary foods, shows contradictory results in the literature [12,13]. A better understanding of environmental factors and critical periods in the life cycle is needed.

For example, birth weight is associated with childhood and adult obesity [14]: a high birth weight (>4000 g) increases the risk of obesity [15], contributing to later childhood obesity and increased body fat mass [16]. While the relationship between low birth weight (<2500 g) and obesity is less clear [17], it is associated with lower lean body mass and greater central adiposity in adults [13], due to rapid catch-up growth in early childhood [13]. Exclusive breastfeeding (the first six months, with continuation and addition of complementary foods until two years or more) [18] is associated with a lower risk of overweight, childhood obesity, and some non-communicable chronic diseases in adulthood [19], possibly due to hormonal differences compared to formula feeding. The inconsistency in the results of previous studies on breastfeeding and obesity [20,21], possibly due to confounding factors, highlights the need for further research, particularly in regions with a high prevalence of excess weight in the child population, such as Mexico.

Regarding prematurity and the risk of childhood obesity, evidence suggests a higher risk of obesity and metabolic problems in preterm infants; however, different studies present inconsistent results regarding the magnitude of this risk [22]. The complex relationship between parental educational level and the development of overweight and childhood obesity shows an inconsistent association, revealing a correlation between low parental education and a higher risk of childhood obesity [23].

Therefore, the objective of this study was to explore the influence of breastfeeding, birth weight, gestational age, parental education, and sex on body mass index (BMI) during infancy.

## 2. Materials and Methods

### 2.1. Participants

Of 310 selected clinical records, 286 corresponding to children aged 2 to 5 years (140 girls and 146 boys) who met the selection criteria between January 2009 and December 2019 were analyzed. Information confidentiality was ensured through anonymization and coding. The study was approved by the Ethics Committee of the School of Dietetics and Nutrition of the ISSSTE (2021.002 L, EDN ISSSTE).

The childcare center has the necessary infrastructure and care to promote exclusive breastfeeding for up to six months and complementary feeding for up to two years, strengthening the mother-child bond and discouraging the use of bottles. The staff of the ISSSTE’s Child Welfare and Development Center Number 26 (EBDI-26) recorded the children’s weight, height, and age every six months, from their admission to the center until age five. This information, recorded in the clinical history, was obtained through weight and height measurements. All personnel received training in standardized WHO techniques before joining the daycare [24]. The family medical history form included questions about birth weight, gestational age, complications during pregnancy and delivery, type of delivery, child’s sex, and neonatal illnesses. Information was also collected on breastfeeding and its duration, the introduction of solid or semi-solid foods, and the family’s environmental and socioeconomic characteristics. In addition, personal data, personal history (pathological and non-pathological), family history, and physical examination were recorded.

Inclusion criteria were as follows: (1) children aged 2 to 5 years; (2) attendance at EBDI-26 between 2009 and 2019; (3) complete and valid information on sex, age, height, and body weight, according to WHO recommendations for the collection, analysis, and reporting of anthropometric data in children ≤5 years [25]; (4a) complete information on the duration of exclusive and total breastfeeding; (4b) complete information on the age of transition to breast milk substitutes; and (4c) complete information on the introduction of complementary feeding and birth weight. Data from children with chronic illnesses that significantly affected anthropometric parameters were excluded, such as congenital heart disease, malabsorption syndromes, hormonal deficiencies, children of women with gestational diabetes and macrosomia, or other similar conditions.

### 2.2. Study Design

A retrospective cohort study was conducted through the review of all clinical records from the 2009–2019 period. A non-probabilistic consecutive sampling of cases that met the selection criteria was selected. Information on the children’s sociodemographic characteristics (sex, age, weight, and height) was obtained. Regarding mothers, information on family income, educational level, and smoking habits was collected and categorized. The WHO standard [24] was used to define overweight and obesity in preschool children.

#### 2.2.1. Nutritional Status of Children

The body mass index (BMI/Age) was calculated using the WHO growth curves for children under 5 years of age to obtain BMI/Age percentiles [26]. Very low weight was defined as a value below the 3rd percentile, low weight between the 3rd and 15th percentiles, normal weight between the 15th and 85th percentiles, overweight between the 85th and 96.9th percentiles, and obesity as values equal to or greater than the 97th percentile. Children who did not have biologically plausible values (i.e., a body mass index/z-score between −5 and +5) were excluded from the analyses.

#### 2.2.2. Breastfeeding

Information on breastfeeding was obtained from the clinical records of the participating children. Information on the initiation, duration, and type of breastfeeding received was collected. For the analysis, the following operational definitions were used:

Exclusive breastfeeding: Defined as the exclusive breastfeeding of the infant without solid foods, formula milk, water, juices, teas, or other liquids during the first six months of life, except for oral rehydration solutions or drops/syrups of vitamins, minerals, or medications. This definition is based on WHO recommendations [27]. The number of months of exclusive breastfeeding was also recorded.

Partial breastfeeding: Defined as infant feeding that includes breast milk but also other liquids (such as water, juices, and infusions), solid or semi-solid foods (purees, papillas), or infant formula.

Breastfeeding patterns were categorized as follows for analysis:

Partial breastfeeding: Time in months of breastfeeding, even with other liquids or infant formula.

Duration of exclusive breastfeeding: Time in months in which the baby was exclusively fed breast milk from their mother and was classified as follows: (1) never exclusively breastfed; (2) exclusively breastfed for less than 6 months (1–5 months); (3) exclusively breastfed for at least 6 months.

This classification was based on the duration and exclusivity of breastfeeding, according to the operational definitions described previously.

#### 2.2.3. Introduction of Complementary Feeding

Information on the introduction of complementary feeding was collected from the clinical records. The introduction of complementary feeding was defined as the introduction of cereals, fruit or vegetable purees, or any other semi-solid or solid food to the child’s diet, in addition to breast milk or infant formula. The age (in months) of the introduction of complementary feeding and the food with which it was initiated were recorded.

#### 2.2.4. Characteristics at Birth

Characteristics at birth were classified according to gestational week and birth weight. Deliveries occurring before 37 weeks of gestation were considered preterm, while those of 37 weeks or more were considered term. Birth weight was categorized according to the 2007 WHO parameters [26]: low birth weight (<2500 g), normal (2501–4000 g), and high birth weight (≥4001 g). This variable was analyzed continuously and categorically. Information on birth weight was obtained from birth certificates provided by parents or guardians before the child’s enrollment in the daycare.

#### 2.2.5. Sociodemographic Variables

The children’s sex and age, as well as the socioeconomic level (expressed in times of the minimum wage, TMW, in Mexico), marital status, occupation, and educational level of the mother, were obtained from the clinical records. The socioeconomic level was categorized according to the median family income, obtained from the parents’ or guardians’ payslips, recorded in the social records. Years of schooling were categorized as follows: primary and incomplete secondary education as basic education (6–8 years of schooling); complete high school and incomplete university studies as middle education (9–16 years of schooling); and university and postgraduate studies as higher education (≥17 years of schooling). Marital status was categorized as married or single.

### 2.3. Statistical Analysis

Descriptive analysis, including comparisons of means, medians, and frequencies, was performed. The statistical significance of differences in frequencies, means, and medians across BMI percentile changes from birth to 2 and 5 years of age was assessed using the Rao-Scott corrected Pearson χ^2^ test. For continuous variables, the Kolmogorov–Smirnov test determined the distribution. Medians were calculated if the *p*-value was less than 0.05; otherwise, means and standard deviations were reported. Other continuous variables, such as family income (9.2 TMW), were categorized according to their medians. Repeated measures ANOVA analyzed changes over time within each sex. Bivariate correlations and linear regression models identified variables explaining changes in BMI percentiles from birth to 2 and 5 years of age, adjusting for maternal age, childcare duration, mode of delivery, and parity. Data were collected in an Excel 2010 spreadsheet (Microsoft Corp., Redmond, WA, USA) and analyzed using SPSS version 23 for Windows (SPSS Inc., IBM Corp., Armonk, NY, USA). A *p*-value < 0.05 indicated statistical significance.

## 3. Results

This study reviewed 500 clinical records of children who attended state public daycares over the past 10 years. However, only 286 contained complete data for analysis. Table 1 presents the maternal characteristics: mean age at delivery of 36 years; 52% with family incomes above 9.2 minimum wages; 89% married with secondary or higher education; and a primary role as caregivers. The average duration of children’s stay at the daycare was 32 months. The maternal age was 36 years, with a range of 18 to 39 years.

Table 1 shows that 80% of participants were born at term (≥37 weeks of gestation), and 68% by cesarean section. Exclusive breastfeeding lasted a median of 5 months. Complementary feeding, which consisted mainly of fruit and vegetable purees, was also initiated around 5 months. Specifically, 52% of infants began the introduction of solid foods between 4 and 5 months, while 48% did so at 6 months.

Figure 1 illustrates a decreasing trend in the prevalence of very low and low weight and an increasing trend in the prevalence of overweight and obesity over time. From ages 2 to 5 years, a significant decrease in the frequencies of low and very low weight is observed, along with an increase and maintenance of overweight and obesity.

Only 23% of the children received exclusive breastfeeding for at least six months. In comparison, 85% of the children received breast milk at some point, while 68% of the preschoolers received partial breastfeeding; it was also observed that 15% did not receive breast milk. The change in BMI percentile from birth to 5 years of age was similar in the exclusive (67%) and partial (71%) breastfeeding groups, *p* = 0.603.

Regarding family income, children who maintained or decreased their BMI between birth and 2 years of age differed from those who increased it; however, this difference was not statistically significant (*p* = 0.055). In contrast, a statistically significant difference (*p* = 0.001) was observed in the change in BMI percentile between birth and 5 years of age according to gestational age, with children who maintained or decreased their BMI percentile differing significantly from those who increased it (see Table 2).

Compared to boys, girls had a significantly lower birth weight (*p* = 0.009), which was associated with a lower BMI percentile at 5 years of age (*p* = 0.008). Significantly longer exclusive breastfeeding was also observed in girls (*p* = 0.006), as well as a greater decrease in BMI percentile between 2 and 5 years of age (*p* = 0.005) (see Table 3).

The analysis revealed a significantly greater increase in BMI percentile in preterm infants compared to term infants [median (interquartile range): 45 (26–75) vs. 35 (9.2–58), *p* = 0.012] (see Table 4).

Furthermore, months of exclusive breastfeeding, maternal educational level, and birth BMI explained 54% of the variance in BMI percentile from birth to two years of age (*p* = 0.001). Similarly, months of exclusive breastfeeding, maternal educational level, gestational age, and sex explained 9.6% of the variance in BMI percentile from two to five years of age (*p* = 0.001). In contrast, gestational age, birth BMI, and sex explained 53.2% of the variance in BMI percentile from birth to five years of age (*p* = 0.001; see Table 5). Finally, at age five, the prevalence of overweight or obesity was significantly higher in boys (35%) than in girls (22%) (*p* = 0.013). However, sex did not influence the trend of percentile change; only time did (see Figure 2).

## 4. Discussion

This study analyzed early factors influencing the body mass index (BMI) of preschoolers. Between birth and two years of age, exclusive breastfeeding, birth BMI, and the mother’s educational level explained 54% of the variability in BMI percentile. Between two and five years of age, these factors, along with gestational age and sex, explained 9.6% of the variability. Considering the entire period (birth to five years), gestational age, birth BMI, and sex explained 53.2% of the variability. At five years of age, the prevalence of overweight and obesity was significantly higher in boys than in girls, with an average obesity prevalence of 10%.

These figures for overweight and obesity in Mexican preschoolers are concerning, reflecting a global trend observed in countries such as the Bahamas, Brunei, Chile, French Polynesia, Kuwait, Guatemala, Puerto Rico, Saudi Arabia, the United States, and the United Arab Emirates [28]. In our sample, the prevalence of overweight (18.5%) exceeded the global average (5.7%), as well as the figures for East Asia and the Pacific (7.8%), Latin America and the Caribbean (7.5%), North America (9.1%), Central and Eastern Europe (6.6%), and Europe and Central Asia (7.9%) [29]. While the prevalence of obesity (10%) exceeded that of Latin America and the Caribbean (9.2%), it was lower than that of Central, Eastern Europe, and Central Asia (10.9%) [30]. The prevalence of overweight and obesity in Mexican preschoolers is significantly high, exceeding global and regional averages in several cases, which underlines the urgency of implementing prevention and control strategies for childhood obesity in Mexico, considering the global trend toward an increase in these figures.

### 4.1. Early Life Factors

#### 4.1.1. Breastfeeding

Our study demonstrates that exclusive breastfeeding for at least six months prevents childhood overweight and obesity, corroborating previous findings [20,21,31]. The results support the hypothesis that exclusive breastfeeding reduces this risk, especially during the first two years of life; children exclusively breastfed for at least six months had a significantly lower BMI. These findings are consistent with other research indicating exclusive breastfeeding as a protective factor against childhood overweight and obesity [32,33,34]. A meta-analysis of 25 studies (more than 226,000 participants) showed a 22% reduction in the risk of obesity in breastfed children, and an analysis of 17 studies showed a dose–response effect: longer duration of breastfeeding, lower risk of obesity [35]. However, Ma et al. [36] found no association, attributing it to the exclusive use of BMI as an indicator. This discrepancy highlights the need to use multiple indicators in future research [37].

Despite the high prevalence of breastfeeding in our study, only 23% received exclusive breastfeeding during the first six months, a figure lower than global rates (44%), those of Latin America and the Caribbean (37%), South Asia (57%), Europe and Central Asia (41%), North America (26%), East Asia, and the Pacific (31%) [38], and even that of Mexico (34.2%) [39], and well below the WHO target (50% in 2025 and 70% in 2030) [40]. Furthermore, only 42% continued with mixed breastfeeding, and at 12 months only 6.5% were still being breastfed. These low rates reflect slow progress and marked international disparities. The high prevalence in Central Asian and African countries is attributed to national interventions, such as support from healthcare professionals [41,42,43].

Inadequate adherence to WHO infant feeding guidelines negatively impacts the health of Mexican children [44,45]. Optimal nutrition in early childhood is crucial for cognitive and physical development, immunity, and the prevention of chronic diseases [44,45]. Several factors contribute to the low rates of breastfeeding in Mexico: the poor implementation of the Baby-Friendly Hospital Initiative and its Code; the lack of knowledge and support from healthcare professionals; high rates of pre-lacteal feeding; the lack of adequate spaces in workplaces; insufficient support for breastfeeding mothers [46,47,48,49]; and the aggressive marketing of infant formula, especially targeting vulnerable mothers [50].

#### 4.1.2. Birth Body Mass Index

Birth weight is another factor associated with the development of childhood overweight or obesity, as demonstrated by our study and growing scientific evidence [51,52]. A correlation is observed between birth weight and BMI percentile at age 5. Numerous studies, including systematic reviews and meta-analyses, demonstrate a significant association between high birth weight (>4000 g) and a higher risk of obesity [17], contributing to an increase in body fat mass during childhood and obesity in adulthood [37,53,54]. Conversely, low birth weight, often the result of adverse conditions during pregnancy, has a less direct and still under-researched relationship with obesity [55]. While the association is not as consistent as with high birth weight, a trend towards lower muscle mass and higher body fat, particularly in the central area, is observed [56]. It is hypothesized that this is due to accelerated “catch-up growth” in early childhood, where children with low birth weight accumulate fat to reach an adequate weight [57,58]. In summary, both extremely high and low birth weights can influence the risk of obesity throughout life, although the relationship with low birth weight requires further investigation.

#### 4.1.3. Gestational Age

We observed that preterm infants had a higher BMI percentile at age 5 than term infants. This finding is consistent with previous studies that reported a higher risk of childhood obesity, central adiposity, and metabolic syndrome in preterm infants [59,60,61]. However, the relationship between prematurity (gestational age) and the risk of childhood obesity has yielded inconsistent results. While some studies show a similar risk in preterm and term newborns [62], others report a higher risk [63,64], and still others associate prematurity with higher childhood body fat [65,66].

Although the exact mechanism by which a rapid increase in BMI percentile in preterm infants leads to childhood overweight or obesity is not yet fully understood, several hypotheses have been proposed, including perinatal programming [67], genetic and nutritional factors, and parental feeding practices [68,69]. Evidence suggests that breastfeeding in infants with rapid weight gain is associated with a lower percentage of body fat in the long term [70], possibly due to adiponectin in breast milk, with its insulin-sensitizing and anti-inflammatory effects [71]. Conversely, formula or mixed feeding is associated with greater weight gain (from birth to 3 years) and a higher BMI later (1–5 years) [72]. Furthermore, higher nutrient intake and accelerated postnatal growth are linked to higher metabolic risks in these children [73].

Therefore, exclusive breastfeeding during the first year of life and the gradual introduction of complementary foods rich in fruits and vegetables are presented as effective strategies for the prevention of obesity in preterm infants [74]. In summary, prematurity increases the risk of obesity and long-term metabolic problems. Prevention requires establishing healthy growth patterns from birth, with careful monitoring of weight gain during infancy and early interventions, including dietary and lifestyle recommendations, guided by healthcare professionals to minimize the risk of obesity and its complications.

#### 4.1.4. Sex and Gender Differences in Childhood Obesity

Sex is another factor associated with the development of overweight and obesity in childhood. Our findings, consistent with international literature [75,76], reveal a higher prevalence of overweight and obesity in boys than in girls. This trend, observed in most high- and upper-middle-income countries among children and adolescents aged 5 to 19 years, is confirmed by several studies. For example, Abarca-Gómez et al. reported that, in children aged 5 to 9 years, a higher prevalence of obesity was observed in boys in 123 of 188 countries (65%); this difference persisted in 112 of 188 countries (60%) for the 10- to 19-year-old group [3,8]. While predominant in high- and upper-middle-income countries, this difference is not observed in low- and lower-middle-income countries [77]. In 44 of 88 high- and upper-middle-income countries, the prevalence of obesity in boys was almost double that of girls in the same age group, as illustrated in countries such as Singapore, Denmark, and Canada [77]. This disparity in obesity prevalence between boys and girls has been previously reported. Canadian data consistently show a higher prevalence of obesity in boys (3–19 years) [78], including a prevalence twice as high of severe obesity in boys (5–9 years) [79]. Similarly, in China, the prevalence of childhood obesity increased constantly, being systematically higher in boys (7–18 years) [77]. In Poland, the prevalence of overweight and obesity increased in children (3–19 years), with a higher incidence in boys [80]. Despite these evident differences in obesity prevalence by sex, little has been debated about their possible causes and implications. Research considering sex and gender variables could reveal crucial findings that might otherwise go unnoticed [75].

Childhood obesity prevalence differs between sexes due to a complex interplay of biological and sociocultural factors. From the fetal stage, girls exhibit slower growth in the final trimesters of pregnancy and, after birth, present a higher percentage of body fat and lower lean mass, resulting in reduced caloric needs [81]. Sex hormones influence this body composition, with significantly higher leptin levels in females, directly linked to fat mass and its production [82]. This is further compounded by potential genetic differences that explain some of the variation in body composition [83]. Brown adipose tissue, with its role in obesity prevention, represents another potential factor, although its impact in childhood requires further research, given its greater presence in adult women.

Concurrently, sociocultural influences contribute to this disparity. In high-income countries, girls tend to consume healthier diets and show greater concern about their weight, reflecting stereotypes that idealize female thinness [75,84]. This perception affects parenting and eating habits, with parents more concerned about their daughters’ weight. Beyond diet, other factors such as access to healthcare, social assistance, sleep duration, physical activity, and screen time also influence this difference [85]. However, the lower prevalence of obesity in girls in developed countries contrasts with the impact of these risk factors, requiring further research for a complete understanding of the phenomenon.

Effective prevention of childhood obesity requires an approach sensitive to sex and gender differences. Influences contributing to childhood obesity manifest early and span multiple levels, including family and community. Instead of focusing solely on the child’s individual weight, prevention strategies should consider the impact of sex and gender. While family interventions focusing on parenting and the home food environment exist, a specific focus according to sex or gender is lacking. Furthermore, the effectiveness of school interventions varies according to gender [86]. To promote healthy habits in both boys and girls, prevention strategies must be sensitive to these differences, adapting interventions to modify dietary behaviors and perceptions of weight according to gender. This includes programs aimed at parents, educating them about healthy weights and appropriate eating patterns for their children according to their sex [87].

#### 4.1.5. Parental Education

This study reveals that low parental education constitutes an early life factor associated with the development of overweight and childhood obesity in children aged 2 to 5 years, a finding consistent with observations in high-income countries and supported by two systematic reviews [88,89]. However, a previous review by Wu et al. did not find this association, possibly due to the economic heterogeneity of the countries included in their analysis [90]. In contrast, in children aged 0 to 2 years, a positive association was observed between parental education and childhood overweight/obesity, similar to that reported in developing countries [91]. This discrepancy could be attributed to the intense advertising of processed infant foods, high in calories and low in nutrients, which replace breastfeeding and healthy complementary feeding. Therefore, it is crucial to strengthen nutritional education for parents and caregivers, accompanied by resources and support to facilitate access to nutritious foods [92]. This finding highlights the complexity of the relationship between parental education, sociocultural context, and child health, since while higher parental education is associated with lower childhood obesity in Western countries, an inverse relationship is observed in many Asian countries [92,93]. In general, parental education significantly influences health knowledge and parenting practices, promoting healthy lifestyles in families with more educated parents [94,95]. Finally, national-level food policies play a fundamental role in the availability and quality of food for the population [92].

#### 4.1.6. Family Income

Childhood obesity carries negative consequences for adult health [96], so its prevention must begin in childhood [97], a crucial stage for future health. Risk factors include family income (high or low), parental obesity, sedentary lifestyle, excessive consumption of foods high in carbohydrates and fats, and low adherence to a balanced diet [98]. In this study, the difference in parental socioeconomic level between children with and without overweight or obesity was borderline and not significant, possibly due to the sample size. To prevent overweight and obesity, healthy diets should be promoted, and advertising of food and beverages aimed at infants and preschoolers should be restricted [99].

#### 4.1.7. Complementary Feeding

Regarding complementary feeding, nearly 95% of the babies in the study received solid, semi-solid, or soft foods between 4 and 6 months of age, differing from WHO recommendations [40]. Although exclusive breastfeeding delays the introduction of complementary feeding and could reduce the risk of being overweight, some studies report lower protein and energy intake in breastfed children compared to formula-fed children [19]. However, in this study, the average age of introduction was not associated with a higher BMI percentile at 2 and 5 years. This could be attributed to the introduction of fruits and vegetables in portions and frequency supervised by the nursery’s medical staff, preventing excessive energy consumption.

### 4.2. Strengths and Limitations of the Study

This study presents several methodological strengths. Its longitudinal cohort design, conducted in a daycare center with health personnel who actively support breastfeeding, minimizes bias from external factors and strengthens internal validity. The homogeneity of the sample, excluding mothers with obstetric complications and children with pre-existing illnesses or neonatal intensive care, allows for a more precise evaluation of the impact of breastfeeding, gestational age, sex, birth weight, and maternal education level on BMI at two years of age. These methodological characteristics increase the reliability of the observed association between the studied factors and preschool overweight/obesity.

However, the study presents limitations. The absence of data on maternal BMI at the time of delivery is a significant limitation, given its known impact on newborn weight. The observed correlation between exclusive breastfeeding and a lower risk of obesity at two and five years does not guarantee a persistent effect throughout childhood, especially in the Mexican context. Despite adjusting for multiple variables, other unmeasured factors that could influence the results cannot be completely ruled out. The sample, selected from a daycare center in Mexico City, limits the generalizability of the findings nationally. Finally, the use of retrospective data from medical records introduces a possible recall bias in the information on breastfeeding. Future studies with larger and more nationally representative samples, including maternal BMI and more robust data collection methods, are necessary to confirm and expand these findings.

### 4.3. Perspectives and Future Research Lines

Future research should delve deeper into the biological and environmental mechanisms that cause childhood obesity, including fetal programming and epigenetics, and evaluate the impact of breastfeeding and other types of feeding. Large-scale intervention studies, sensitive to cultural and socioeconomic context, are needed to improve breastfeeding rates and promote healthy eating habits. Improved evaluation measures are required, utilizing more comprehensive indicators than BMI and considering sex and gender differences when designing interventions. Finally, it is crucial to develop and evaluate nutritional education interventions for parents and caregivers.

## 5. Conclusions

This study demonstrates that several early factors significantly influence the BMI of preschoolers. Exclusive breastfeeding for at least six months is associated with a lower BMI at five years of age, corroborating its protective role against childhood overweight and obesity. However, the rates of exclusive breastfeeding in this study were considerably lower than WHO recommendations, highlighting the need for interventions to improve adherence to infant feeding guidelines. Birth weight also showed a correlation with BMI at five years, with both very high and very low weights associated with a higher risk of obesity. Prematurity was associated with a higher BMI percentile at five years, although the relationship is not consistent in the literature. Finally, a higher prevalence of overweight and obesity was observed in boys than in girls at five years of age, highlighting the importance of considering sex differences in prevention strategies. Low parental education was associated with a higher risk of obesity in children aged 2 to 5 years, highlighting the need to strengthen nutritional education for parents and caregivers. Taken together, these findings emphasize the importance of multifactorial interventions aimed at promoting breastfeeding, controlling birth weight, addressing the specific needs of preterm infants, considering sex and gender differences, and promoting parental nutritional education to prevent childhood obesity.

## Figures and Tables

**Figure 1 nutrients-17-00697-f001:**
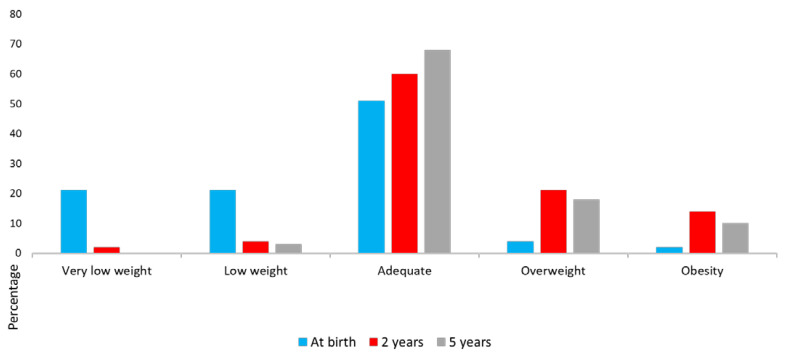
BMI (%) from birth to 5 years, assessed at three time points. Significant differences were observed in BMI between birth and 2 years (*p* = 0.001). No significant differences were found between birth and 5 years (*p* = 0.507) or between 2 and 5 years (*p* = 0.725). These analyses were performed using the Pearson χ^2^ test.

**Figure 2 nutrients-17-00697-f002:**
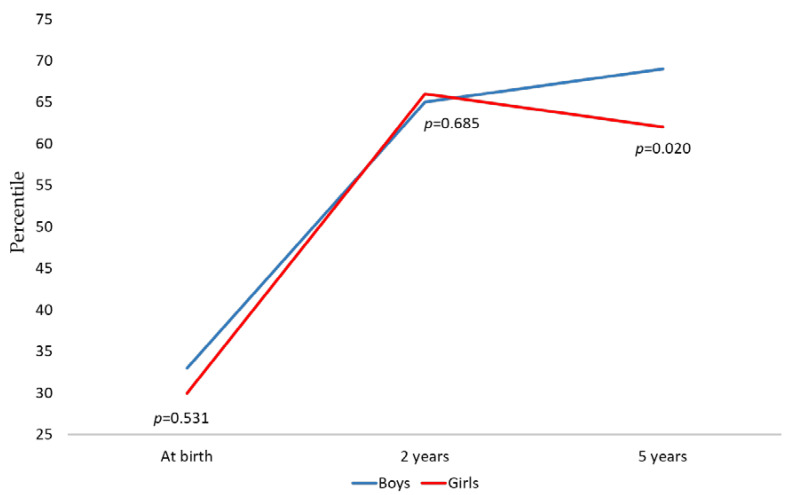
BMI percentile trends from birth to 5 years of age, stratified by sex. Within-subjects analysis showed a significant effect (*p* = 0.001). There was no significant effect of sex between groups (*p* = 0.238). These results are based on repeated measures ANOVA.

**Table 1 nutrients-17-00697-t001:** General characteristics of mother–preschool child dyads (*n* = 286).

Characteristics	Frequency (%)
Socioeconomic characteristics	
Family income (TMW)	
≤9.2	137 (48)
≥9.2	149 (52)
Marital status	
Single	36 (13)
Married	250 (87)
Occupation	
Professions	211 (74)
Housekeeper	18 (6)
Officious	57 (20)
Mother’s education level	
Basic education	30 (11)
Secondary education	107 (37)
Higher education	149 (52)
Gender of caregiver	
Man	28 (10)
Woman	258 (90)
Characteristics of the baby	
Mode of delivery	
Vaginal delivery	92 (32)
Cesarean delivery	194 (68)
Months of exclusive breastfeeding	
Age at introduction of complementary feeding (months)	5 (2–8) *
First food introduced for complementary feeding	5 (2.5–6) *
Fruits	176 (61.5)
Vegetables	110 (38.5)
Gestational age (weeks)	37.6 (31–42) *
Gestational age	
Preterm	56 (20)
Term	230 (80)
Sex	
Girl	140 (49)
Boy	146 (51)

^(^*^)^ Median (25th percentile–75th percentile), TMW: times the minimum wage.

**Table 2 nutrients-17-00697-t002:** Distribution of values for different variables according to changes in BMI from birth to 2 years old (a), from 2 to 5 years (b), and from birth to 5 years old (c).

Change	Loss or Without Change	Increased	* *p*
Birth to 2 years	*n* = 116	*n* = 170	
Gestational age	38 (37–39)	38 (37–39)	0.492
Family income (TMW)	8.3 (5.7–13.4)	9.8 (6.7–14.1)	0.055
Maternal age (years)	33 (29–36)	32 (29–35)	0.285
Complementary feeding beginning (months)	5 (4–6)	5 (4–6)	0.958
Exclusive breastfeeding (months)	4 (3–6)	4 (2–5)	0.399
Breastfeeding (months)	6 (3–8)	5 (2–7)	0.061
BMI at birth	57 (23–80)	11 (1.5–25)	0.001
BMI from 2 to 5 years	*n* = 237	*n* = 49	
Gestational age	38 (37–39)	38 (36–39)	0.166
Family income (TMW)	9.3 (6.4–14.7)	86 (5.5–11.6)	0.071
Maternal age (years)	32 (29–35)	33 (29–36)	0.953
Complementary feeding beginning (months)	5 (4–6)	5 (4–6)	0.473
Exclusive breastfeeding (months)	4 (3–5)	4 (0.5–6)	0.678
Breastfeeding (months)	5 (3–7)	6 (0.5–9)	0.561
BMI at birth	20 (6–53)	23 (2–55)	0.835
Birth to 5 years	*n* = 138	*n* = 148	
Gestational age	38 (38–39)	38 (37–39)	0.001
Family income (TMW)	8.6 (6.1–13.4)	9.7 (6.7–14.3)	0.215
Maternal age (years)	33 (30–36)	32 (29–35)	0.186
Complementary feeding beginning (months)	5 (4.4–6)	5 (4–6)	0.485
Exclusive breastfeeding (months)	4.2 (3–6)	4 (2–5)	0.232
Partial breastfeeding (months)	5.5 (3–7)	5 (2–8)	0.797
BMI at birth	55 (21–76)	10 (1–21)	0.001

TMW: times the minimum wage; * data expressed as median (25th percentile–75th percentile); *p*-value by Mann–Whitney U tests.

**Table 3 nutrients-17-00697-t003:** Comparison of infant characteristics by sex.

Variable	Female	Male	* *p*
Complementary feeding (month)	5 (4.5–6)	5 (4–6)	0.379
Birth weight (g)	3000 (2685–3285)	3175 (2800–3450)	0.009
BMI at birth (percentile)	18.6 (4–55)	22 (10–54)	0.192
BMI at 2 years (percentile)	72(47–91)	72(39–91)	0.856
BMI at 5 years (percentile)	65 (43–82)	74 (54–93)	0.008
Gestational age (weeks)	38 (37–39)	38 (37–39)	0.691
Exclusive breastfeeding (moths)	5 (3–6)	4 (1–5)	0.006
Breastfeeding (months)	5 (3–6)	4 (1–7)	0.009
Family income (TMW)	9.2 (6.1–14.4)	9.5 (6.6–13.5)	0.773
Change at birth to 2 years old	37.5 (10–68)	36 (8–59)	0.486
Change from 2 to 5 years old	−3 (−21–10)	2 (−11–10)	0.005
Change from birth to 5 years old	39 (12–64)	34 (8–59)	0.211

TMW: times the minimum wage. * *p*-values are from Mann–Whitney U tests.

**Table 4 nutrients-17-00697-t004:** General characteristics of infants according to gestational age.

Variable	Term (*n* = 230)	Preterm (*n* = 56)	* *p*
Complementary feeding (month)	5 (4–6)	5 (4–6)	0.951
Birth weight (g)	3100 (2810–3400)	2830 (2300–3200)	0.001
BMI at birth (percentile)	21 (7–59)	13 (0–46)	0.008
BMI at 2 years (percentile)	72 (44–91)	68 (38–95)	0.912
BMI at 5 years (percentile)	70 (55–89)	69 (46–87)	0.205
Exclusive breastfeeding (months)	4 (3–5)	3 (0–6)	0.191
Breastfeeding (months)	6 (3–8)	4 (0–8)	0.072
Family income (TMW)	9.2 (6.5–13.7)	9.1 (5.5–13.2)	0.726
Change at birth to 2 years old	36 (7–62)	37 (11–81)	0.307
Change from 2 to 5 years old	−0.2 (−17–15)	1.6 (−13–20)	0.219
Change from birth to 5 years old	35 (9.2–58)	45 (26–75)	0.012

TMW: times the minimum wage. * *p*-values are from Mann–Whitney U tests.

**Table 5 nutrients-17-00697-t005:** Linear regression models predicting the preschooler’s BMI in percentiles.

Variable	Mean	B	Coefficient	R	R^2^ Adj ^1^	(*p*)
BMI change from birth to 2 years	34.3					
Constant		32.675	-	0.744	0.540	0.001
Months of exclusive breastfeeding		−0.738	−0.087			0.039
Month of complementary feeding beginning		0.402	0.011			0.807
Family income (MS)		0.028	0.050			0.271
Educational level		3.832	0.099			0.027
Gestational age		0.587	0.027			0.512
BMI at birth		0.950	0.712			0.001
Sex (1 girl, 2 boys)		1.693	0.021			0.604
BMI change from 2 to 5 years	0.073					
Constant		108.525		0.349	0.096	0.001
Months of exclusive breastfeeding		0.987	0.154			0.009
Month of complementary feeding beginning		−1.130	−0.039			0.515
Family income (MS)		−0.018	−0.148			0.507
Educational level		−4.388	−0.151			0.017
Gestational age		−2.312	−0.143			0.015
BMI at birth		0.081	0.080			0.170
Sex (1-girl, 2-boy)		−8.921	−0.148			0.010
BMI change from birth to 5 years	34.4					
Constant		141.2	-	0.545	0.532	0.001
Months of exclusive breastfeeding		0.249	−0.032			0.447
Month of complementary feeding beginning		−0.728	−0.021			0.629
Family income (MS)		0.010	0.020			0.662
Educational level		0.555	−0.016			0.725
Gestational age		−1.725	0.088			0.037
BMI at birth		0.869	0.716			0.001
Sex (1 girl, 2 boys)		−7.229	−0.100			0.016

Change in BMI was in percentile. ^1^ Adjusted for maternal age, daycare duration, mode of delivery, and parity.

## Data Availability

The data presented in this study are available on request from the corresponding author. The data are not publicly available due to privacy restrictions.

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
