# Peer review of "Early Life Determinants of Overweight and Obesity in a Sample of Mexico City Preschoolers"

_nutrients, 2025, doi:10.3390/nu17040697_

Round 1
Reviewer 1 Report
Comments and Suggestions for Authors
I had the great privilege to review the manuscript entitled “Early Life Determinants of Overweight and Obesity in Mexico City Preschoolers.” However, the manuscript can be improved by addressing the following points that the authors should consider:
1. Introduction: The background discussion is overly lengthy, with a disproportionate focus on global obesity issues, while insufficient emphasis is given to the specific context of childhood obesity in Mexico and the study’s core value. The literature review should be more targeted, with a clear research hypothesis or objective, and should highlight the study's novelty.
2. Materials and Methods: This section lacks sufficient detail regarding the study design, particularly how biases were controlled and data accuracy was ensured (e.g., addressing potential biases in parent-reported breastfeeding data). The rationale for statistical methods and strategies for handling missing or extreme data values is not fully explained, which affects the transparency and credibility of the results. The representativeness of the sample, being limited to Mexico City, is not adequately discussed, potentially limiting the generalizability of the findings. Additionally, the exclusion criteria for participants, such as those with chronic conditions affecting anthropometric measures, should be explicitly detailed to ensure clarity and strengthen the validity of the results.
3. Results: Figure 1 lacks statistical validation and can only describe trends, reducing its scientific value. Statistical testing should be added to strengthen the credibility of the figure. Additionally, L229-L251 presents overly dense data, which may overwhelm readers. The authors should focus on key conclusions and avoid redundant details. For example, when describing the relationship between gestational age and BMI percentile changes, it would be more effective to directly highlight that BMI increases significantly more in preterm infants compared to term infants (p = 0.012) and briefly explain its significance rather than reiterating table data. The link between statistical results and study conclusions should also be clarified, such as explaining that gestational age and birth BMI are major predictors of BMI changes.
4. Discussion: The discussion is overly lengthy and lacks structure, with repetitive and poorly organized interpretations of the results. It should more systematically compare the study findings with existing literature and emphasize the study’s contributions and innovation. Contradictory findings, such as the relationship between breastfeeding and BMI at different ages, require deeper analysis. Specific public health recommendations, such as interventions for preterm infants and gender-specific strategies for obesity prevention, should also be provided.
5. Figures and Tables: Figure 1 lacks statistical validation and is purely descriptive. The visual design should be improved for better clarity, and table content should be simplified to enhance readability and data communication.
Author Response
Reviewer 1
I had the privilege of reviewing the manuscript entitled "Determinants of Overweight and Obesity in Early Childhood in Preschool Children in Mexico City." However, the manuscript can be improved by addressing the following points:
- We thank Reviewer 1 for their valuable observations, which have significantly improved our manuscript.
- Introduction: The background discussion is too extensive, with a disproportionate focus on global obesity problems, while insufficient emphasis is given to the specific context of childhood obesity in Mexico and the study's central value. The literature review should be more specific, with clear research hypotheses or objectives, and should highlight the study's novelty.
- Response: In the revised version, the introduction has been restructured to focus on the specific context of childhood obesity in Mexico, including associated early childhood factors.
- Materials and Methods: This section lacks sufficient detail on the study design, particularly regarding how biases were controlled and data accuracy was ensured (for example, addressing potential biases in parent-reported breastfeeding data). The justification of the statistical methods and strategies for handling missing or extreme data values are not fully explained, affecting the transparency and credibility of the results. The representativeness of the sample, limited to Mexico City, is not adequately discussed, potentially limiting the generalizability of the findings. Furthermore, the exclusion criteria for participants, such as those with chronic diseases affecting anthropometric measurements, should be explicitly detailed to ensure clarity and strengthen the validity of the results.
- Response: The "Materials and Methods" section underwent significant revision to improve transparency and rigor.
- Study Design: A complete description of the study design was provided, including the study type (retrospective cohort), sampling process, and bias control strategy.
- Statistical Methods: An explanation of the statistical tests used was included.
- Sample Representativeness: The limitation of the sample to Mexico City, along with geographical limitations and their impact on external validity was explained.
- Inclusion/Exclusion Criteria: Inclusion and exclusion criteria were clearly defined, including a detailed explanation of why participants with chronic diseases affecting anthropometric measurements were excluded.
- Results: Figure 1 lacks statistical validation and can only describe trends, reducing its scientific value. Statistical tests should be added to strengthen the figure's credibility. Furthermore, lines 229-251 present overly dense data, potentially overwhelming readers. The authors should focus on key conclusions and avoid redundant details. For example, when describing the relationship between gestational age and changes in BMI percentiles, it would be more effective to directly highlight that BMI increases significantly more in premature infants compared to term infants (p = 0.012) and briefly explain its importance instead of reiterating the table data. The link between the statistical results and the study's conclusions should also be clarified, explaining how gestational age and birth BMI are the main predictors of BMI changes.
- Response: The Results section was improved to optimize the presentation and impact of the findings.
- Figure 1: Statistical tests were included in Figure 1 to support the observed trends, and the p-value was added.
- Conciseness and Clarity (Lines 229-251): The information was simplified, focusing on key findings.
- Link between Results and Conclusions: The connection between the statistical results and the study conclusions is now explicit and clear, explaining in more detail the main predictors of BMI changes. The Results section is now more concise, focusing on the most relevant results and establishing a clear and logical connection between the presented data and the study's conclusions.
- Discussion: The discussion is too long and lacks structure, with repetitive and poorly organized interpretations of the results. It should compare the study's findings with existing literature more systematically and emphasize the study's contributions and innovations. Contradictory findings, such as the relationship between breastfeeding and BMI at different ages, require further analysis. Specific public health recommendations should also be provided, such as interventions for premature infants and gender-specific strategies for obesity prevention.
- Response: The discussion was made concise and better structured, providing a critical and reflective analysis of the findings, including a rigorous comparison with existing literature, a clear identification of the study's contributions and limitations, and the formulation of specific and actionable public health recommendations.
- Figures and Tables: Figure 1 lacks statistical validation and is purely descriptive. The visual design should be improved for greater clarity, and the table content should be simplified to improve readability and data communication.
- Response: The figures and tables were improved to increase clarity and informational value.
- Figure 1: The figure was revised to include statistical tests supporting the observed trends, and the p-value was added. The visual design was also improved to ensure clarity, conciseness, and ease of interpretation.
- Tables: The table content was simplified to improve readability.

Reviewer 2 Report
Comments and Suggestions for Authors
Dear Authors,
Thank you for the opportunity to review this manuscript. The manuscript addresses an important public health issue by analyzing potential predictors of overweight and obesity from very early stages of ontogeny. Given that the study population is at high risk for overweight and obesity, the findings have significant potential for application. Despite the authors' use of strict inclusion criteria and the resulting homogeneity of the data, there are a few limitations to the study.
Comments on the manuscript, in the order they appear in the text:
1. The description of the material (e.g., educational level categories) in the text differs from the education categories presented in Table 1. Please clarify and explain these discrepancies.
2. The table header should indicate that percentages are provided in parentheses within the table.
3. Is the range for TMW 9.8 (5.7-1.4) (page 6 of 18, second line from the bottom) correct?
4. Given the assumption that "A p-value < 0.05 was considered statistically significant," p=0.055 is not significant (page 6 of 18, second line from the bottom).
5. In Table 5, the characteristics should be listed in the same order.
6. It is necessary to include available socioeconomic data, such as at least the level of education of the caregiver/mother/father, in the linear regression model. The level of education is a well-known basic determinant of health literacy and generates differences in health status, including body weight, in children (e.g. https://pubmed.ncbi.nlm.nih.gov/31924589/). Parental education has a determining influence on children's growth processes (both at the exogenous and epigenetic levels). It is necessary to include this variable in the analysis as a potential determinant and to mention it in the theoretical chapters of the manuscript.
Kind regards,
reviewer
Author Response
Reviewer 2
Thank you for the opportunity to review this manuscript. The manuscript addresses an important public health issue by analyzing potential predictors of overweight and obesity from very early stages of ontogeny. Given the study population's high risk of overweight and obesity, the findings have significant potential for application. Despite the authors' use of strict inclusion criteria and the resulting homogeneity of the data, the study has some limitations.
Comments on the manuscript, in the order they appear in the text:
- Reviewer 2 provided valuable observations that we have incorporated into the revised manuscript.
- The description of the material (e.g., educational level categories) in the text differs from the education categories presented in Table 1. Please clarify and explain these discrepancies.
- Response: The educational level categories have been harmonized in the text and Table 1.
- The table header should indicate that percentages are provided in parentheses within the table.
- Response: Table 1 header was modified to indicate that percentages are provided in parentheses.
Is the range for TMW 9? 8 (5.7-1.4) (page 6 of 18, second line from the bottom) correct?
- Response: The error has been corrected.
- Given the assumption that “a p-value < 0.05 was considered statistically significant”, p = 0.055 is not significant (page 6 of 18, second line from the bottom).
- Response: The manuscript now specifies that only p-values < 0.05 are considered significant.
- In Table 5, the characteristics should be listed in the same order.
- Response: The error has been corrected; they are now listed in the same order.
- It is necessary to include available socioeconomic data, such as at least the caregiver/mother/father's educational level, in the linear regression model. Educational level is a well-known basic determinant of health literacy and generates differences in health status, including body weight, in children (e.g., https://pubmed.ncbi.nlm.nih.gov/31924589/). Parental education has a determining influence on children's growth processes (both exogenously and epigenetically). It is necessary to include this variable in the analysis as a potential determinant and mention it in the theoretical chapters of the manuscript.
- Response: Caregiver's educational level was included in the statistical analysis as a potential determinant. Its importance was discussed in the introduction, and the results were analyzed and contrasted with published information (e.g., https://pubmed.ncbi.nlm.nih.gov/31924589/). Furthermore, the theoretical discussion on the influence of parental education on childhood growth, both exogenously and epigenetically, has been expanded.

Round 2
Reviewer 1 Report
Comments and Suggestions for Authors
The V2 version has been improved, but further refinement in data presentation, enhanced explanation of statistical methods, and more specific contributions and practical implications would enhance the manuscript’s academic impact.
- Results: In Figure 1, the authors aim to illustrate a decreasing trend in very low and low weight and an increasing trend in overweight and obesity over time. However, the placement of p-values does not clearly indicate whether the comparisons refer to overall trends or specific categories. It is recommended to use the numbers (or percentages) for the five weight categories as the dependent variable and age groups as the independent variable to perform a trend analysis for clearer presentation. Similarly, Figure 2 focuses on BMI changes by sex, and applying the same approach would improve the clarity and interpretability of the results.
- Discussion: The discussion section still repeats statistical results from the results section instead of focusing on interpretation. For example, the results already state that at age 5, the prevalence of overweight or obesity is 35% in boys and 22% in girls. Instead of reiterating these figures, the discussion should focus on potential explanations for this gender difference.
Author Response
Reviewer 1-Round 2
The V2 version has been improved, but further refinement in data presentation, enhanced explanation of statistical methods, and more specific contributions and practical implications would enhance the manuscript’s academic impact.
Thank you for your valuable feedback. We appreciate your suggestions regarding the presentation of data, statistical methods, and the discussion of specific contributions and practical implications. We will carefully consider these points during our revisions.
Results: In Figure 1, the authors aim to illustrate a decreasing trend in very low and low weight and an increasing trend in overweight and obesity over time. However, the placement of p-values does not clearly indicate whether the comparisons refer to overall trends or specific categories. It is recommended to use the numbers (or percentages) for the five weight categories as the dependent variable and age groups as the independent variable to perform a trend analysis for clearer presentation. Similarly, Figure 2 focuses on BMI changes by sex, and applying the same approach would improve the clarity and interpretability of the results.
Response: Figures 1 and 2 have been revised. Figure 1 now presents the counts (percentages) of each weight category across age groups, along with statistical analysis showing trends across these categories. Similarly, Figure 2 now displays the changes in BMI according to sex using an appropriate statistical analysis to clarify the observed trends.
Discussion: The discussion section still repeats statistical results from the results section instead of focusing on interpretation. For example, the results already state that at age 5, the prevalence of overweight or obesity is 35% in boys and 22% in girls. Instead of reiterating these figures, the discussion should focus on potential explanations for this gender difference.
Response: The discussion section has been revised to reduce the emphasis on statistical reiteration and to better focus on interpreting the observed gender difference in overweight/obesity prevalence at age 5, exploring potential contributing factors.
Sincerely,
Professor Hugo Martínez Rojano
School of Medicine
Mexico City

Reviewer 2 Report
Comments and Suggestions for Authors
Dear Authors,
I thank the authors for accepting the comments and making corrections to the manuscript. It has now acquired a higher quality.
I will add that the indicated article proves definitely stronger connections between the parent's level of education and the child's biological condition, because it concerns irreversible, unidirectional growth processes (body height). It therefore constitutes a lifelong investment of the parent in the child and their health awareness. The manuscripts cited by the authors, despite their obvious advantages, concern a reversible feature (body mass / BMI).
Reviewer
Author Response
Reviewer 2-Round 2
I thank the authors for accepting the comments and making corrections to the manuscript. It has now acquired a higher quality.
I will add that the indicated article proves definitely stronger connections between the parent's level of education and the child's biological condition, because it concerns irreversible, unidirectional growth processes (body height). It therefore constitutes a lifelong investment of the parent in the child and their health awareness. The manuscripts cited by the authors, despite their obvious advantages, concern a reversible feature (body mass / BMI).
Thank you for your insightful comment highlighting the distinction between irreversible (height) and reversible (BMI) characteristics. We appreciate your feedback.
Sincerely,
Professor Hugo Martínez Rojano
School of Medicine
Mexico City
